# Situation-Aware IoT Data Generation towards Performance Evaluation of IoT Middleware Platforms

**DOI:** 10.3390/s23010007

**Published:** 2022-12-20

**Authors:** Shalmoly Mondal, Prem Prakash Jayaraman, Pari Delir Haghighi, Alireza Hassani, Dimitrios Georgakopoulos

**Affiliations:** 1School of Science, Computing and Engineering Technologies, Swinburne University of Technology, Hawthorn 3122, Australia; 2Department of Human-Centred Computing, Monash University, Clayton 3800, Australia

**Keywords:** IoT, IoT middleware platforms, benchmarking, Fuzzy Situation Inference, IoT data generation, situation transition, performance evaluation

## Abstract

With the increasing growth of IoT applications in various sectors (e.g., manufacturing, healthcare, etc.), we are witnessing a rising demand of IoT middleware platform that host such IoT applications. Hence, there arises a need for new methods to assess the performance of IoT middleware platforms hosting IoT applications. While there are well established methods for performance analysis and testing of databases, and some for the Big data domain, such methods are still lacking support for IoT due to the complexity, heterogeneity of IoT application and their data. To overcome these limitations, in this paper, we present a novel situation-aware IoT data generation framework, namely, SA-IoTDG. Given a majority of IoT applications are event or situation driven, we leverage a situation-based approach in SA-IoTDG for generating situation-specific data relevant to the requirements of the IoT applications. SA-IoTDG includes a situation description system, a SySML model to capture IoT application requirements and a novel Markov chain-based approach that supports transition of IoT data generation based on the corresponding situations. The proposed framework will be beneficial for both researchers and IoT application developers to generate IoT data for their application and enable them to perform initial testing before the actual deployment. We demonstrate the proposed framework using a real-world example from IoT traffic monitoring. We conduct experimental evaluations to validate the ability of SA-IoTDG to generate IoT data similar to real-world data as well as enable conducting performance evaluations of IoT applications deployed on different IoT middleware platforms using the generated data. Experimental results present some promising outcomes that validate the efficacy of SA-IoTDG. Learning and lessons learnt from the results of experiments conclude the paper.

## 1. Introduction

The rise in IoT applications is unparalleled in the last few years [1]. IoT middleware cloud platforms are used extensively for IoT deployments as they provide many functionalities such as managing IoT devices, real-time data processing, distributed data storage, among others. This complicates the already strenuous comparison of IoT middleware platforms, making it essential to have a standard solution that enables IoT application developers to compare different IoT middleware options best suited for their application. A key step in developing such a solution is to be easily conduct performance evaluation of IoT middleware platforms running the IoT applications. One key step in being able to achieve this is to be able to generate IoT data that is close to real-world data to undertake performance evaluation of IoT application running on different IoT middleware platforms and identify performance bottlenecks. Such data for IoT scenarios for evaluating the performance of IoT middleware platforms can either be real-world data from various real-world domains of IoT, which require access to large volumes of data generated by the IoT devices, or they can be simulated data that has been artificially generated which mimics real-world data, with known underlying patterns. While using real data in performance evaluation in not always recommended [2], it is difficult to replay the data sequentially with the real frequency. Additionally, real data may not always be available to represent real-world events such as earthquakes or road accidents. However, developers need access to data to test and debug their application and often rely on synthetic data suited for their needs which often does not properly represent real-world scenarios, which motivates the need for efficient data generation techniques. While domains such as databases have well-established standard datasets [3,4], and Big Data domain has some emerging data sets [5,6,7], IoT still lacks a methodology to generate comprehensive data that can capture the properties of IoT applications (e.g., real-time dynamic data)and hence can be used to evaluate the performance of IoT middleware platforms. To generate data while keeping the significant features of real IoT data, it is important to understand the different needs and requirements for IoT applications. However, relatively little effort has been given to considering the application requirements to generate data. Additionally, data generators in the existing literature do not properly represent the data generated by IoT devices, which are usually in the form of continuous stream of data. One such example is the frequency with which the data should be generated. RIoT Bench [8] for example, addresses the conventional data analytic tasks that are carried out over real-time IoT data streams. However, the data are not representative of event-based data, which is generated in most IoT applications, since IoT applications are mostly event or situation driven.

IoT applications are often associated with real-world situations. The notion of ‘Situations’ is used to represent any real-world events, activities or situations such as ‘road accidents’, ‘fire’, ‘walking’. It has been established from the literature that sensory data combined with contextual information can help in better understanding of the environment [9,10]. With the evolution of IoT, and the increasing use of such IoT applications, it is crucial to have an additional layer of contextual information to generate IoT data corresponding to real-time situations. For example, let us consider the example of an IoT application for real-time monitoring of traffic conditions of a road segment. The situations associated with such an application that describe real-world scenarios can be: ‘low_traffic’, ‘moderate_traffic’, and ‘high_traffic’. Sensory data such as location, speed, density, and trip time, combined with context information such as location, and weather can be used to define the situations. The defined situations can then be used as a baseline to generate real-world IoT data. Moreover, in real-world, the situations gradually transition from one situation to another. For example, a situation of ‘low_traffic’ gradually transition to ‘moderate_traffic’ and then to ‘high_traffic’. Capturing such details to generate data makes data more realistic.

Existing research efforts in IoT data generation have used different methods. Some of them are discussed below: TPCx-IoT [11] benchmark generated data based on the Yahoo Cloud Serving Benchmark framework (YCSB) [12]. The dataset is based on data from sensors from an electric power substation. The authors in [13,14] used machine learning based models to generate time series smart meter data. However, there is a prevalent gap in the literature since data has not been generated considering real-world situations, and their transitions. To address the aforementioned challenges, in this paper, we propose a novel framework, SA-IoTDG, to generate data that is similar to real-world data and takes into consideration the IoT application requirements, the real-world situations and their transitions. Hence, rather than randomly generating data without considering the IoT application requirements, we aimed to address the need for simulating situation aware data for IoT applications. Two key components of proposed SA-IoTDG are: (1) a data generation tool for generating sensor data, keeping the significant features of real data, and (2). A SySML model that captures the requirements of IoT applications. This model is an extension to our past work [15]. We extend this model to capture the IoT application situations and the transitions. Our proposed framework allow users to define their application entities, the situations for their application, and generate IoT data. This enables them to perform simulations prior to having the actual dataset. In addition, most IoT applications, rely on supervised learning algorithms, where, training the dataset is essential. The accuracy of these algorithms depends highly on the training and test process. It is very difficult to find a comprehensive training dataset that gives high accuracy. Hence, in this paper, we use a Fuzzy Logic based situation theory approach to generate relevant IoT data that does not require training the dataset.

Our approach to defining situations for IoT applications and generating data to test and evaluate IoT middleware platforms running IoT applications is based on Fuzzy Situation Inference (FSI) [16], which combines Fuzzy Logic and Context Spaces theory. While Fuzzy Logic provides the flexibility for reasoning and considers the uncertainties and inaccuracies of a situation for an IoT application, Context Spaces theory provide an additional layer of contextual information to provide enhanced reasoning about real-world situations. In [16], the authors state the usage of FSI for situation inference under uncertainty. The situation transitions are handled by a Markov chain-based model integrated into the framework. The proposed framework can help both researchers and IoT application developers to generate IoT data for their application. The generated dataset can be pushed to different middleware platforms to compute relevant benchmarks. The tool is extensible, and can be configured and tailored to user specific needs and requirements. The innovation of the system lies in its ability to deal with the uncertainty and transition between different situations and generate data in near-real-time that reflects the dynamic properties of IoT data.

The contributions of the paper are as follows:We propose a novel situation based data generation framework which serves as a starting point for identifying performance bottlenecks of IoT deployments. The framework is configurable and can be extended for multiple IoT use cases.We extend the IoTSySML model [15] to represent IoT situations and transitions.We introduce a Markov chain-based approach for situation transition to generate real-world IoT data to support performance evaluation of IoT middleware platforms running IoT applications.We evaluate the data generation capability of the framework by simulating a traffic monitoring application that validates the dynamic data generation as situation transitions occur.

The outline of this paper is as follows: Section 2 gives an overview of the related work. Section 3 introduces the basic concepts required for this work. Section 4 presents the proposed framework for situation aware data generation. Section 5 demonstrates the implementation of the proposed framework. Section 6 presents the experimental evaluations and the analysis. Finally, Section 7 concludes the paper.

## 2. Related Work

As discussed above, IoT middlewares are being used extensively in IoT. However, there is limited literature in IoT Middleware benchmarks and how the different middlewares can be compared with each other. The most notable work in this area is of [17,18] where the authors have compared two publish-subscribe based middlewares, FIWARE and M2M, based on a set of qualitative and quantitative metrics. The middleware platforms have been considered as a black box without considering the internal implementation. Other works [19,20] that compare middleware platforms are limited to conventional metrics such as reliability, scalability, price, and availability. However, the complexity of IoT applications gives rise to the need for metrics tailored for IoT domain. For example, Medvedev et al. [21], evaluated the performance of OpenIoT platform from data ingestion and data storage perspective. The authors in [22] use contextual queries for benchmarking IoT context management platforms. In another research by Salhofer et al. [23], the different components (General Enablers) of the FIWARE platform has been thoroughly analyzed and evaluated from an application deployment point of view. The authors reported that IoT application could benefit by utilizing the various services offered by the platform. However, the lack of updated documentation makes the process of application development difficult. Cruz et al. [24] presents a performance evaluation study of a few existing middleware platforms and establish qualitative and quantitative metrics to evaluate the performance of the middleware platforms, such as InatelPlat, Konker, Linksmart, Orion+STH and Sitewhere. There have also been a few works on evaluating the performance of Fiware platform to investigate its usability in various domains of IoT. For example, Araujo et al. [25] evaluated the performance of FIWARE with respect to scalability to report its usage in smart city applications. Similarly, another study [26] was carried out to explore the applicability of FIWARE platform in the smart farming domain. Table 1 summarizes the related work in this area. From the table, we can see that benchmark studies in IoT are fragmented. Existing research in this area targets the different layers of the IoT ecosystem. For instance, ref. [27] is a device level benchmark for IoT edge devices. Similarly, while TPCx-IoT [28] targets the IoT gateways, RIoTBench [8] evaluates the stream processing capabilities for IoT applications. However, it is limited to DSPS applications that are centrally hosted in the cloud. In addition, ref. [29] targets cloud-based applications. Hence, while standard holistic benchmarks exist in other related areas such as big data [5,7,30,31,32], databases [3,4], RDF stream processing [33,34], the same cannot be said for IoT.

It can be established from the literature [5,31,32,33,36,37] that a dataset is an important factor for conducting performance evaluation of a system. However, there is limited work around real world data generation in IoT targeted for performance evaluation of IoT middleware platforms. Some of the prominent works in the area are IoTSim [38] which aims to analyse IoT applications, but is limited to Mapreduce models. The other notable work is of Cardoso et al. [17] where the dataset used is data from a traffic monitoring scenario that publishes the average speed of traffic in each street of the city on an hourly basis. Another research [39] proposed a dataset that targets Intrusion Detected Systems for IoT and IIoT application domain. Iftikhar et al. [14] used a predictive based model to generate time series smart meter data. Similarly, Liu et al. [13] generate smart meter datasets using a regression model. There are a few simulators that have been used in the IoT space, most of which are applicable to cloud and Wireless Sensory Network (WSN). Full stack simulators such as Cloudsim [40], the OASIS standard Devices Profile for Web Services (DPWSSim) [2], and iFogSim [41] have been used to simulate IoT application scenarios. iFogSim is limited to fog and edge computing. Simulators such as IoTSim [38] and SimIoT focus on the data processing of IoT applications. While IoTSim is restricted to simulating MapReduce programming model, SimIoT is currently unable to simulate different types of sensors. Simulators such as OMNET++ [42], Cooja [43], NS2 [44] are network simulators and have been widely used in WSN. However, these simulators hardly provide support for performance evaluation in IoT.

As stated earlier, our key objective is to use the generated data for evaluating the performance of different IoT middleware platforms. So, we want data to be generated that mimics real-data. To achieve this, we might take into consideration the requirements of IoT applications and other details, for example, real-world situations that they are associated with. For example, in a fire monitoring application, the severity of a fire can be classified into situations such as ‘low threat’, ‘medium threat’ or ‘high threat’ and actions can be taken accordingly. We have reviewed the literature to investigate the methods used for modelling and representing situations of IoT applications. In [45], the authors propose a domain specific language for representing IoT applications. Some of the works based on context modelling technique have defined situations using a form of predicate calculus [46]. Pre-defined rules are triggered when a situation occurs and situations are inferred based on the result set ‘true’, ‘false’ or ‘possibly true’. Other context modelling approaches focused on the temporal aspects of a situation [10,47]. This work was further extended to address the uncertainty aspect of IoT [48]. Other works leveraged Fuzzy Logic-based approaches to represent the uncertainty and dynamic situations of IoT applications [49,50]. However, just using Fuzzy Logic to represent IoT applications was not enough, and not suitable for IoT environments. Hence, it can be seen in the literature that Fuzzy Logic has been integrated into Context Spaces theory as a formal method of representing situations for ubiquitous environment, which is more suitable for IoT applications [16,51]. IoT applications based on real-world situations can benefit using FSI modelling, as it enables generating data according to the application requirements. Additionally, FSI captures the minor and gradual changes which occurs in real-world situations. As an example, in the fire monitoring application discussed above, if the fire threat is ‘low’, the fire situation can be handled by just sending a warning message, whereas if the fire threat is ‘high’, actions to put out the fire and an alarm should be issued. In this way, the false alarm rate can be decreased. Without an FSI approach, there are no intermediate values, but absolute values: Is there a fire? yes/no or 0/1. Based on the FSI modelling technique, a set of rules are used to define Situations. These FSI rules are then triggered to infer a situation. Multiple rules can be used to define a situation [16]. However, most of these existing approaches do not consider the transition between these situations. In real-world situations, gradually transition from one state to another. If we take the same traffic monitoring scenario described above, a situation of low_traffic gradually transitions to moderate_traffic. Capturing these granular details enables generating a more realistic, looking dataset. Based on the literature, Markov chain is one of the most common approach that has been used to handle state transitions [52,53]. Some approaches [54,55,56] combine Markov chain and Fuzzy Logic to take into consideration the uncertainties while computing the transition probability matrix by replacing the crisp transition probabilities by fuzzy numbers.

In summary, benchmarks in IoT are under researched and there is limited research work around generation for evaluating the performance of IoT middleware platforms. Table 1 illustrates that none of the existing studies take into consideration the IoT application requirements in generating data that is similar to real world data. IoTAbench [36] is the closest to our work, which addresses our goal partially. The authors have developed a Markov chain-based synthetic data generator for their framework. While the data generation approach is interesting and generates realistic time-series data, the data generator does not consider IoT application requirements to generate relevant data. We have highlighted in Table 1 how our proposed framework will address the existing gaps in the literature including (a) devising a template for specifying IoT application requirements, (b) generate IoT data taking into consideration application’s requirements. Furthermore, to the best of our knowledge, there is no data generator in the IoT domain which considers IoT situations and their transitions for generating IoT data. The IoT data simulator is a more recent attempt [57], that generates data on the fly while also having the capability of replaying existing data with modified data. However, it does not provide support for simulating situation based IoT data, generating data for multiple situations or simulating the situation transitions.

## 3. Preliminaries

In this section, we discuss some of the concepts used for this research for representing situations and generating relevant IoT data.

### 3.1. Fuzzy Situation Inference Theory

The Fuzzy Situation Inference (FSI) Theory [16] can be used for representing and inferring real-world situations. FSI builds on the theory of Context Spaces [10] which is a formal approach for context modelling and reasoning context-aware applications. To overcome the limitations of this theoretical modelling approach, in dealing with uncertainty of real-world situations, the FSI approach combines Context Spaces with Fuzzy Logic-based approach to benefit from both approaches in one methodology. We will first give a brief overview of the Context Spaces Theory (CST) and then discuss the FSI theory.

The Context Space Theory (CST) [10] was developed as a formal approach for reasoning about context in multidimensional space, known as Situations. Some other basic concepts of CST are described below:**situation space**, where real-world situations are perceived as sub-spaces within a *n* dimensional application space.**context attributes**, which are the data generated from sensors that can be used for reasoning or inferring situations. Context attributes can be either measured by sensors directly, or derived from sensory data. For example, air temperature, light level, noise level, air humidity, etc., can be the context attributes for a smart office scenario.**region** which is a domain of allowed values for a context attribute.**context state** which represents a data point within an *n* application space at a point of time, where *n* is the number of context attribute.

The term situation is commonly used to represent a contextual state in an application space. There exists several real-world situations in the IoT application space, such as ‘meeting’ in a smart office scenario [48] or a situation of ‘hypertension’ in a health monitoring scenario [16]. Situation awareness is the specification of such complex real-world situations by taking into consideration its relationship in space and time [58].

In FSI, we use the term ‘linguistic variable’ to represent any contextual parameters. A situation is represented using FSI rules. The rules comprise linguistic variables, each of which constitute a fuzzy set. An FSI rule consists of an antecedent and a consequent, similar to fuzzy rules. The antecedent can be associated with more than one fuzzy set. FSI rules consists of multiple conditions joined by fuzzy operators. The most commonly used are an ‘AND’ or ‘OR’ operators. A single FSI rule has the form:
R01: If CA1 is A and CA2 is B, then the Situation is C
where the *‘if’* part is the antecedent, and the *‘then’* denotes the consequent. *CA1*, *CA2* and *CA3* are linguistic variables defined by fuzzy sets *A*, *B*, and *C*. These rules are evaluated using membership functions. Some of the commonly used membership functions are triangular function, trapezoidal function among others. FSI uses the notion of weights, and confidence to infer a situation [10]. Each variable in the rule is assigned a weight (a value between 0 and 1) which determines the importance of the variable in determining a situation. The consequent of the rule determines the degree of certainty with which a situation is occurring and is given by the below equation:(1)$$Confidence=\sum_{i=1}^{n}w_{i}\mu \left(x_{i}\right)$$
where $\mu (x_{i})$ is the membership degree of ith element in the fuzzy set *x*, and $w_{i}\mu \left(x_{i}\right)$ is the weighted membership degree of the element $x_{i}$. A situation is said to occur if its degree of certainty tends to 1. We will discuss this in details in Section 4.3.

### 3.2. Markov Chain

In conditional probability theory, a Markov chain $X_{n},n\geq 0$ is a stochastic model, whose state space *S* is a finite set, for T=(0,1,2,⋯). $X_{n}$ denotes the state of the system at time *n*. It can be used to calculate the probability of moving from one state to another and is expressed as $X_{n}=i$, which denotes $X_{n}$ being in state *i*. Markov chains are characterized by the Markov property, which states that the transition to a state at time *t*+1 depends only on the current state at time *t*. Figure 1 illustrates a generic Markov chain.

For any $j,x_{0},x_{1},\cdots ,x_{n}$, the Markov property is defined by:(2)$$P\left(X_{n+1}=j|X_{0}=x_{0},\cdots ,X_{n}=i_{n}\right)=P\left(X_{n+1}=j|X_{n}=i\right\}$$

The above equation (Equation (Equation 2)) emphasizes that the probability of $X_{n+1}$ depends only on the probability of $X_{n}$. The idea is that given the present state, the past states have no influence on the future state or simply put, the future state depends only on the present state. Hence, the Markov chain is ‘memory less’, since the probability of transitioning to a different situation depends on the current situation, but not on the past situations. The multitude of these probabilities to transition from one state to another is represented in a matrix form, called a transition probability matrix *P*, where $P=[P_{ij}]$, where $P_{ij}$ is the probability of moving from state *i* to state *j*. The rows represent the current state and the columns represent the next state. The probability of $X_{n+1}$ being in state *j*, given that $X_{n}$ is in state *i*, is denoted by $P_{i}^{n}j^{n+1}$, i.e.,
(3)$$P_{i}^{n}j^{n+1}=Pr\left\{X_{n+1}=j|X_{n}=i\right\}$$

The above equation explains that the transition probabilities are not only a function of initial and final states, but also of the time of transition.

## 4. Situation Aware IoT Data Generation Framework

This section presents the SA-IoTDG (Situation Aware IoT Data Generation) framework that can be used to define situations, simulate the transition between the situations and generate IoT data based on the order of the occurrence of the situations. In the following subsections, we start with a high-level system overview, followed by the key components of the proposed framework.

### 4.1. System Overview

As shown in Figure 2, the system broadly comprises SA-IoTDG, our proposed framework (gray dotted box), communication channels, and IoT middleware platforms to run IoT applications. Users provide their application requirements ➀ to the framework to generate IoT data. The generated data can be sent to different cloud managed IoT middleware platforms over various communication channels ➆.

*SAIoT-DG Architecture: *SAIoT-DG comprises a user interface ➁ to register the situations and run simulations, IoTSySML_X ➄, a domain reference model to capture user specific application requirements, components to model the situations ➃ and their transitions ➄, and a component to generate the data ➅. We describe briefly the key components for developing the SA-IoTDG framework. The framework is divided into four major parts:*IoTSySML_X:* We extend on our previous IoTSySML model [15] which was developed to capture the requirements of IoT applications. We integrate the concepts of situations and context from the domain of CST [10] to strengthen the model.*Situation Description System:* We define real-world situations of IoT application using an FSI [16]-based approach.*Situation Transition Model:* We handle the various transitions between the situations. The transitions have been modelled using Markov Chain concepts [53,59].*IoT Data Generation:* We focused on generating data with respect to the defined situations. The generated data should mimic the real-time IoT data as much as possible. Another essential feature we wanted was to enable users give their desired configuration. By configuration, we mean a set of parameters that controls the data being generated. An example is a parameter to declare the frequency of the data generation.

In the following subsections, we discuss each of these components in details.

### 4.2. IoTSySML_X

In this section, we extend on our previous model [15] to enhance its functionality. Our previous model IoTSySML [15] was developed to provide a formal way to capture the requirements of IoT applications, which was missing in existing literature. Identifying and modelling the application requirements enables generating data relevant to the IoT applications. For this paper, we have extended this model to include situation theory, Context Spaces and Fuzzy Logic concepts. By adding these new elements, we are providing support for more context to represent an IoT application scenario and their requirements in a better way. For example, we represent IoT applications, using situations from real-world scenario. If we take the same example of the IoT traffic monitoring scenario discussed above, the application can be represented by situations of ‘high_traffic’, ‘low_traffic’, or ‘moderate_traffic’. Adding contextual information such as location (e.g., parking areas), weather conditions (e.g., snow or rain), or road conditions (e.g., road work, accidents) provides granular details which enable realistic data generation for evaluating the performance of IoT middleware platforms. The model has been conceived using a UML class, where we extend existing UML concepts using Stereotypes. A concept is encapsulated using a ‘block’. Each block is a stereotyped (customized) extension of a UML Class and has a set of properties and defined relation with other blocks. We first explain the extension we applied to the traditional UML. Figure 3 shows the stereotypes that we used for the model. The stereotype ***‘Context’*** has been extended from existing UML model. The ‘Context’ class represents the context information of any IoT application scenario and is defined by elements *‘ContextSpace’*, *‘ContextualParameter’*, and *‘ContextElements’*. The ContextSpace stereotype represents the situations or events a context element might be involved in. ContextElement describes the entity itself in a situation. A context element can be a user, or any physical object. ContextualParameter represents the observable properties from an IoT device. Similarly, the stereotype ***‘Situation’*** has been extended from existing UML and entails all information related to defining a situation and is defined by stereotypes *‘SituationProperty’* and *‘FSIElements’*. SituationProperty represents the characteristics of a situation. For example, the transitions a situation might go through. FSIElements represent the conditions applied to defining a situation.

We now illustrate the class diagram of the domain concepts depicted in Figure 4. The green elements represent concepts from our previous model [15]. The purple boxes indicate the extended concepts. The *‘IoT devices’* element is the virtual representation of a physical IoT device. IoT device class comprises *‘Sensor’*, which makes *‘Observation’*, a class provided by SOSA [60] to capture a sensor observation. It represents the data generated by the sensors and have attributes: *‘Phenomenon Time’* to capture the temporal aspect of an observation as a window or in intervals, and *‘Values’* to represent the observations made by the sensors (for example, acceleration values generated by accelerometer or temperature values generated by temperature sensors). Values are represented as a key-value pair. A geolocation observation for example has two key value pairs of latitude, longitude and can have a type which can have value such as circle. Every sensor is associated with *‘metadata’* [61], which has attributes type and value. An example of metadata to represent the unit of a temperature sensor data is: ‘type’ = “string’, ‘value’ = “Celsius’.

*‘Entities’* are a representation of any physical object or individual, and are involved in *‘Situations’*. ‘Situations’ are any real-world situation such as ‘low traffic’ or any activity such as ‘working’. They are characterized by ID, name, timestamp, definition, previous state and certainty (probability with which a situation is occurring). An example of an entity ‘worker’ being involved in a situation would be (id=$S_{1}$, name = ‘cutting meat’, prev_state = ‘idle’, certainty = 0.7). A situation comprises a set of attributes involved in defining the situation at a particular time. These attributes have been modelled as a *‘Context’* class and refers to characteristics or information associated with ‘Sensor Observation’. It also entails data and behavior of the entity class [48]. Hence, we see a relationship between Entity and Context. Additionally, we can observe that every instance of ‘Context’ has an association relation with *‘Variables’* which refers to characteristics or information associated with ‘Observation’. Table 2 summarizes the relationships between the different classes in the model. Variable are stereotyped as ‘ContextualParameters’ and are characterized by name, definition, weight (the relative importance of the parameter in inferring a situation) and is associated with a *‘FuzzyRule’* element stereotyped as *‘FSIElelements’*. Another FSIElement is the element Fuzzy Rule, which is involved in defining a situation. A situation can be defined using multiple ‘Variables’. Similarly, there can be multiple FSIRule instances to define one instance of any situation. The *‘Transition’* element allows the situation class to move from one state to another and is stereotyped as a feature of the situation class. To allow the transitions, three types of triggers were considered: time-constraint, rule based, and external environmental factors. Each of these has been captured by the *‘Triggers’* class.

### 4.3. Situation Description System

In this section, we describe our situation description system. The components of the system are a fuzzifier, a fuzzy inference engine, a rule base, and a situation reasoner. Figure 5 illustrates these components. We explain the process of FSI based situation definition, with our traffic monitoring motivating scenario.

#### 4.3.1. Fuzzifier

Fuzzification entails defining the input and output parameters to the FSI system to determine the traffic condition of a road segment. The fuzzifier receives the crisp context information as input from the sensors. In FSI, these crisp inputs are converted to a fuzzy set by the fuzzifier. Fuzzy sets are stipulated by their membership functions, $\mu_{A}\left(x\right)$ which determined how an input is mapped to membership values between 0 and 1 given by Equation (Equation 4)
(4)$$x_{k}=\mu_{A}\left(x_{k}\right)$$
where $x_{k}$ is the input data and $\mu_{A}\left(x_{k}\right)$ is the is the degree of membership of an element *x* in the fuzzy set *A* for every xϵA.

A traffic monitoring scenario discussed in Section 1 can be involved in situations such as ‘low_traffic’, ‘moderate_traffic’, and ‘high_traffic’. We use three linguistic variables to define these traffic situations. These variables and their fuzzy subsets are described below:

**Speed, α:** Denotes the average speed of the cars on the road segment.
(5)$$\alpha =\left\{\begin{array}{cc} 1 & \mathrm{slow},\;0\geq \alpha \geq 50\\ 2 & \mathrm{normal},\;40\geq \alpha \geq 80\\ 0 & \mathrm{fast},\;70\geq \alpha \geq 120 \end{array}\right.$$

**Density, β:** Average density or the number of cars on a road segment.
(6)$$\beta =\left\{\begin{array}{cc} 1 & \mathrm{low},\;0\geq \beta \geq 15\\ 2 & \mathrm{normal},\;10\geq \beta \geq 35\\ 0 & \mathrm{high},\;30\geq \beta \geq 45 \end{array}\right.$$

**Trip time, γ:** Average time taken to cross the road segment.
(7)$$\gamma =\left\{\begin{array}{cc} 1 & \mathrm{less},\;0\geq \gamma \geq 15\\ 2 & \mathrm{usual},\;10\geq \gamma \geq 30\\ 0 & \mathrm{longer},\;25\geq \gamma \geq 40 \end{array}\right.$$

#### 4.3.2. FSI Inference

The FSI Inference component consists of a rule base and an Inference Engine. It receives the contextual parameters as inputs from the fuzzifier component and uses them to evaluate the FSI rules. A rule base in an integral part of an FSI based inference system. These fuzzy rules are defined by domain experts and are stored in a Rule repository. The rules are subjective and rely on the discernment of domain experts. An example of an FSI rule is shown below:
R01: If A AND B AND C, THEN X
where *A*, *B* and *C* are rule conditions to define a Situation *X*. As an example, the condition *A* can be written as ‘speed is slow’, *B* is another condition ‘density is ’high’. Similarly, the variable *C* can be written as ‘trip_time is longer’. These conditions denote Situation *X* is occurring. To simplify, the same rule can be written as:
R01: If speed is ‘slow’ AND density is ‘high’ AND trip_time is ’longer’, THEN situation is ‘High Traffic’

Table 3 shows a subset of the defined FSI rules for our traffic monitoring scenario.

The inference engine evaluates these pre-defined rules to define a situation. We formulated the rules based on an extensive analysis of IoT traffic monitoring scenario applications [62,63,64,65]. We adopt the Mamdani fuzzy inference algorithm [66] for defining our situations for IoT applications. We have categorized each input variables into different fuzzy subsets. We represent the fuzzy subsets by a trapezoidal membership function. Figure 6 shows that an input variable can belong to two fuzzy subsets simultaneously, with varying degrees of membership. A single xϵX may be associated with multiple fuzzy subsets. As an example, the for the input linguistic variable speed, a crisp value of 45 km/h can either belong to fuzzy subset ‘slow’ or ‘normal’ (Figure 6). The membership function assigns a degree of similarity to these values. If applied to the ‘slow’ fuzzy subset, the membership function might assign a degree of 0.2 ($\mu_{slow}\left(45\right)=0.2$). The same value, when applied to the ‘normal’ fuzzy set, might have a degree of 0.9 assigned ($\mu_{normal}\left(45\right)=0.9$). This signifies, a speed of 45 km/hr has greater chances of belonging to the fuzzy subset normal.

#### 4.3.3. Situation Descriptor

Each situation is associated with a confidence value, which is calculated using:(8)$$Confidence=\sum_{i=1}^{n}w_{i}\mu \left(x_{i}\right)$$

We illustrate the final step for defining the situation ‘Low Traffic’ for the below rule:
R01: If speed is ‘normal’ AND density is ‘less’ AND trip_time is ’less’, THEN situation is ‘Low Traffic’

Table 4 shows the input values of the linguistic variables speed, density and trip_time. Each of these variables have been assigned a weight, which indicates its relevance in defining a situation. For example, we have assigned a weight of 0.4 to ‘speed’ and ‘density’ which denotes both these variables have the same importance in defining the situation ‘low_traffic’. Additionally, these variables have been mapped to fuzzy subsets and assigned a degree of membership by a trapezoidal membership function (Equation (Equation 4)) depicted by the $\mu_{x}\left(i\right)$ column.

We calculate the confidence value below using Equation (Equation 8):



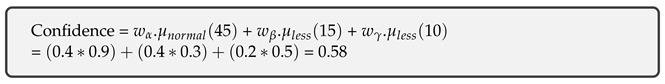



The situation descriptor defines the occurrence of a situation based on this confidence value. From the confidence value, we can say that there is 0.58 probability that the situation Low Traffic is occurring. Hence, the FSI based model discussed above gives a mapping between the inputs (α, β AND γ) and the output (δ) and hence define the situations.

### 4.4. Situation Transition Model

After situations have been defined using FSI, in this section, we present our proposed situation transition model which has been conceptualized using Markov chains [67].

Markov models can be represented by state machines, where the transition of each state can be modeled as a continuous-time Markov process as shown in Figure 7, where $P_{i}$ and $P_{j}$ are the probabilities of transitioning from state i to state j. These transitions between the states generates a sequence of states over time. For example, Figure 7 shows a Markov model with two states, $S=S_{1},S_{2}$. The possible sequence of states could be:



$$\left\{S_{1},S_{2},S_{1},S_{1},S_{1},S_{2},S_{2},S_{1},S_{2},S_{2},\cdots \right\}$$



The probability of transitioning to the next state can be estimated by performing experiments and can be expressed as a transition probability matrix. The transition probability matrix can be determined based on experimental data, such as from a measured sequence of events [56]. In Figure 7, if there is a transition of state $S_{i}$ to state $S_{j}$, the transition probability matrix can be calculated by [56], where
(9)$$\pi_{ij}=\frac{N_{ij}}{N_{oi}}$$

$\pi_{ij}$ is the probability of transitioning from state $S_{i}$ to state $S_{j}$, $N_{ij}$ is the number of observed transitions from state $S_{i}$ to state $S_{j}$, and $N_{io}$ = $\sum_{j=1}^{M}N_{ij}$, is the total amount of data that belongs to state $S_{i}$ i.e., the total number of transitions that are initiated from state $S_{i}$.

We explain our proposed model with our traffic monitoring motivating scenario and define some of the concepts used for this research below:


1.**Situation Space***S* = $\{S_{1},S_{2},\cdots ,S_{n}\}$: corresponds to the possible situations of an IoT application. For our scenario the situation space would be:(a)
*low_traffic*
(b)
*moderate_traffic*
(c)
*high_traffic*
2.**Transition Kernel** = $k(S_{n}|S_{n+1})$: denotes the probability of transition from situation $S_{n}$ to situation $S_{n+1}$ given that situation $S_{n}$ has already occurred, with certain restrictions shown below:
➝k(low_traffic|low_traffic)=p, probability of staying in the same situation, which is semantically equivalent to moving to the same state➝k(low_traffic|moderate_traffic)=1−p, probability of moving to moderate_traffic from low_traffic➝k(low_traffic|high_traffic)=0, no direct transition from low to high_traffic➝k(moderate_traffic|low_traffic)=p, probability of making a transition to low_ traffic to moderate_traffic➝$k\left(moderate\_traf\;fic|moderate\_traf\;fic\right)=1-p^{2},$, probability of staying in the same state of moderate_traffic➝$k\left(moderate\_traf\;fic|high\_traf\;fic\right)=1-p^{2}$, probability of moving to high_traffic from moderate_traffic➝k(high_traffic|low_traffic)=0, no direct transition from high to low_traffic➝k(high_traffic|moderate_traffic)=p, probability of making a transition to high_traffic to moderate_traffic➝k(high_traffic|high_traffic)=1−p, probability of staying in the same high_traffic situation


As seen in the above conditions, we restrict the direct transition of situation low_traffic to a situation of high_traffic or a direct transition from high_traffic situation to low_traffic situation. Simply put, we cannot transition from low_traffic to high_traffic situation in a single step, we need two steps in making the transition. The two-step transition probability is given by matrix $P^{2}$:
(10)$$P\left(S_{2}=j|S_{0}=i\right)=P\left(S_{n+2}=j|S_{n}=i\right)=P_{ij}^{2}$$

By Equation (Equation 2), the two-step probability for our traffic scenario is represented as:
$$P(low\_traf\;fic|high\_traf\;fic)=p(low\_traf\;fic,moderate\_traf\;fic)\;p(moderate\_traf\;fic,high\_traf\;fic)=P^{2}$$


3.**State Space***s* = {InitialState,TransitionState,TargetState}: corresponds to the possible states in a situation transition sequence:(a)*initial state*—corresponds to the starting state for a transition sequence;(b)*transition state*—refers to the intermediate states a situation goes to before transitioning to the target state;(c)*target state*—corresponds to the final state a situation transitions into.4.**Target Triggers**: corresponds to the triggers that might result in a situation transition. For this work, we have considered two triggers:(a)time-based—situations transition according to a time specified by users. After the threshold time is crossed, situation $S_{1}$ transitions to situation $S_{2}$ and continues the current execution, and(b)probability based—based on the probabilities given, situations transition from one state to another5.**Situation Transition Probability Matrix** tPr: denotes the probability with which a situation would move to another. The elements of the transition matrix tPr are defined as:
$${\left(tPr\right)}_{ij}=P_{ij}=P\left(S_{1}=j|S_{0}=i\right)=P\left(S_{n+1}=j|S_{n}=i\right)$$
for any *n*, where $P_{ij}$ is the probability of making a transition from state ‘*i*’ to state ‘*j*’.


Assuming there are *n* situations defined in our Situation Space which are denoted by $S_{1},S_{2},\cdots S_{n}$, and we represent each situation with a state in the Markov model, we will have *n* the number of states. The situations gradually transition from one state to another, i.e., there is a probability of the situations happening in a sequence $P(S_{i},S_{j}\cdots Sn)$. The transition probability from the *i*th to *j*th state is given by a probability transition matrix $P_{ij}$. Hence, a Markov chain having *n* possible states, has a n∗n transition probability matrix given by:$$\begin{array}{c} P=\left[\begin{array}{cccc} P_{11} & P_{12} & \cdots & P_{1n}\\ P_{21} & P_{22} & \cdots & P_{2n}\\ . & . & . & .\\ . & . & . & .\\ . & . & . & .\\ P_{n1} & P_{n2} & \cdots & P_{nn} \end{array}\right] \end{array}$$

Our system contains a finite set of states, and we represent the states by a finite state machine, where the transition of each state has been modeled as a continuous-time, finite-state Markov process. For our IoT traffic monitoring motivating scenario, we assume we have three situations, where each situation is modelled as a Markov state illustrated in Figure 8. However, our framework can be configured to add more situations if required. The user interface developed as part of the framework enables users to add situations as required by their IoT application. The diagram outlines the probabilities associated with making a transition from one situation to another. For example, it shows that there is a 50% chance to move from $S_{3}$ to $S_{2}$, and a 30% chance to transition to $S_{3}$ from $S_{1}$.

Each state or situation in the Markov model has been characterized using Fuzzy Situation Inference Theory, as discussed in Section 4.3. There might be multiple situations occurring at the same time, with varying degrees of confidence. Hence, our proposed approach leverages Markov chains and incorporates fuzzy based reasoning to enhance the prediction accuracy of the situation transitions.

#### Next Situation Prediction

The execution of the situation transition model is given by a Markov model in the form of a deterministic finite automata (DFA) shown in Figure 8. The transitions in a Markov chain are probabilistic rather than deterministic, which means that there is no absolute certainty about what situation will occur at time t+1. Each situation $S_{i}$ is associated with a Markov state and the transition probability matrix determines the probability of transitioning between the states, i.e., between the situations $S_{i},i=1,\cdots ,n$. The transition probability of the occurrence of a traffic situation at time t+1, given the current traffic situation at time *t*, can be written as:(11)$$siTPr=P(S_{t}|S_{t+1})$$

As stated above, we have three situations for our scenario, the transition model for these situations has been illustrated in Figure 8. A Markov chain model can be described completely based on two key factors:

**Transition Probability:** We express the transition probabilities in the form of a transition matrix below. For this experiment, for convenience we assume, equally likely probabilities to transition from one situation to another. Hence, the next situation prediction model is given by
(12)$$S_{n+1}=S_{n}*tPr$$
where tPr = $\left(P_{ij}\right)$ is the situation transition probability matrix given by:$$\begin{array}{c} \mathrm{tPr}=\left[\begin{array}{ccc} P_{11} & P_{12} & P_{13}\\ P_{21} & P_{22} & P_{23}\\ P_{31} & P_{32} & P_{33} \end{array}\right] \end{array}$$

The above transition probability matrix, tPr describes the traffic situation of a road segment and has the following properties:(i)*Property 3.1* The matrix is a stochastic matrix, i.e., $P_{ij}$ in a row should add to 1(ii)*Property 3.2* If $P_{ij}\geq 0$, then the situation $S_{n}$ can transition to $S_{n+1}$

The transition probability matrix defines the probability of transitioning from one situation to another. Hence, the next situation can be determined as a function of the current state and the transition probability, as given by Equation (Equation 12). Additionally, the probability of staying in the same state is semantically equivalent to moving to the same state, which are shown by self loops in Figure 8. Furthermore, if situation $S_{1}$ moves to situation $S_{3}$, and passes another transition state $S_{2}$, the probability of going to $S_{3}$ after n steps can be determined by:(13)$$p\left(S_{1},S_{3}\right)=p\left(S_{1},S_{2}\right)*p\left(S_{2},S_{3}\right)=p^{2}$$

*An Initial State Vector:* This is a probability vector which initializes a Markov chain and denotes the probability distribution of the initial states, i.e., each element in the vector represents the starting probability of a transition sequence at a given state. We initialize our Markov model and denote the initial state probability vector by:
$$P_{initial}\left(p_{1},p_{2},p_{3}\right)$$

The initialization can be conducted in multiple ways, we might assume a fixed starting point, or we can allocate probabilities for starting the Markov process. For our work, we assume equally likely probabilities for starting the sequence of transitions, where our initial state probability vector would be $P_{initial}$ = (0.33,0.33,0.33) since we have three states in our Markov model.

### 4.5. IoT Data Generation

The IoT Data Generation is a key component of SA-IoTDG framework, which handles the data generation aspect. It has been built on the existing IoT data simulator stack [57]. Since, we aim to generate IoT data for the purpose of evaluating IoT middleware platforms, we would ideally want the generated data to be similar to real-world IoT data. Hence, to enable a realistic data generation, we consider the context of IoT applications, so the data generated is more close to real world IoT data. To achieve this, we have extended the functionality of the IoT data simulator to enable data generation for situation aware IoT applications. To elucidate further, as seen in the previous subsections, we define our situations leveraging FSI theory (Section 3.1), and specify the transition between these situations based on a Markov model (Section 4.4). The data generation component receives inputs from these modules: component ➃ and ➄ (Figure 2) and generates data based on information received from these components. Hence, the framework is able to adapt dynamically to the changing configuration, and generate data based on that.

Furthermore, IoT applications might have different requirements in terms of generating data. Some applications might require data to be generated more frequently than others. For example, a smart farming application [68] might require data from soil sensors every few minutes to monitor the soil quality. On the other hand, our traffic monitoring motivating scenario might require data to be generated every few seconds to obtain the real-time location and speed information of vehicles. Hence, rather than randomly generating data, we aimed to address the need for simulating situation aware data for IoT applications by considering the IoT application requirements. As an example, for this work we conceptualized an IoT traffic monitoring scenario, and generated a traffic dataset. The entity here is a road segment under consideration. The generated data mimics data from on road camera sensors in the particular road segment and consists of the average speed of the vehicles on a particular road segment, the vehicle count, and the estimated travel time. The frequency of the generation of the observations can be configured by the user. Furthermore, since most IoT smart city applications are location aware, we generate the data with geolocation coordinates of the users. For our experiment, we have generated observations with a sampling frequency of 1/5 (i.e., observations every 5 s). The data generation component is capable of generating data in near-real-time. We demonstrate the entity, the IoT devices involved in such a scenario, and the data they generate. SSN [69] and SOSA [70] ontologies are often used to represent sensor data observations. Figure 9 shows a sample ontological representation of generated traffic data observations using SOSA ontology. The data generated by the framework correspond to a one traffic data. We generated data with a sampling frequency of 1/5 s, to align it with real-world traffic data.

## 5. Implementation of SA-IoTDG

In this section, we discuss the implementation details of the proposed framework (SAIoTDG) and explain how an application developer can use the framework to generate data relevant to his IoT application using sequence diagrams. SAIoTDG allows users to define situations corresponding to their IoT application with respect to the situation description system discussed in Section 4.3, which is illustrated in Figure 10. SAIoTDG also provides a functionality for users to configure the sequence of the situation transitions, which is illustrated in Figure 11. When the order and occurrence of a known situation is detected, the framework starts generating IoT data for the situation according to the configuration set by the user (sense frequency, number of data records to be generated).

At the beginning, ‘users’ (IoT Application Developers) enter their application requirements to IoTSySML_X which is an extension of our previous work [15]. The SySML model returns an XML specification document of the application requirements, which is first parsed into intermediate JSON and then a template is generated containing all the required details to register a situation. In the next step, the *Situation Manager* validates and stores the situation details in the *Situation Repository*. (Step 2 and 2.1) We used a MongoDB database to store the situation specific details. When a request is made to obtain the situation details (Step 3), a connection is established with the Situation Repository, that connects to MongoDB service, and fetches the situation details as a response in JSON format (Step 3.1).

After successfully, registering the situations, users can specify the situation transition sequence to generate data. The workflow for running the simulation is illustrated in Figure 11. First, users need to specify the order of the situation transitions (Step 1). At this step, the parameters for data generation can be configured (Step 2). Once the desired sequence of transitions and the configurations have been selected, we establish a web-socket communication for generating the data. Parameters can be updated (Step 2.2) and data are generated with the new settings in near-real-time as the situation transitions occur. We use web sockets in FAST API (python web framework) for instantaneous data transfer without HTTP overhead. The generated data are then displayed to the user (Step 2.2.1) and then data can be sent to IoT middleware platforms for further performance evaluation (Step 2.2.1).

*User Interface:* We developed a User Interface (UI) to allow users to generate and visualize data based on their IoT application. The user interface has been developed using react JS and is responsible for: (i) registering new situations for IoT applications, (ii) enabling the transitioning between situation, and (iii) generating data based on defined situations and the configured sequence of situation transitions. The UI consists of different components for each functionality. The ‘Add Situation’ component (Figure 12a,b) enables to register situations and consists of several input fields: application name, context attributes, the fuzzy sets, fuzzy ranges, etc. After the application details have been added, a summarized view of the registered situations can be seen on the application dashboard (Figure 12c).

The situation transition sequence is specified by the ‘Run Situation Transition’ component. It has been developed using ReactJS. Redux has been used to pass data between the components as a centralized way of managing react states in the application. The required APIs for the situation repositories have been developed using NodeJS Express framework for smooth interoperability between the different components. The application can be deployed using docker for seamless integration of the services deployed in different docker container and to enable the application to run in any environment. The application is run using the docker-compose.yaml file. Additionally, Axios library has been used to send HTTP requests to the REST endpoints. The ExpressJs framework has been used to create REST API endpoints, and to request data from the database. It handles the functions to process various CRUD operations. ExpressJs uses mongoose as an Object Data Model library and connects with MongoDB service. Hence, when the request is made from ReactJS to the ExpressJS Rest API, a connection is established and the generated data are displayed to the UI console.

## 6. Case Study and Evaluations

This section presents the evaluation of the proposed SA-IoTDG framework in terms of its ability to generate data based on the defined situations, enable situation transitions and evaluate performance of IoT middleware platforms. We will be using the POC prototype implementation to show the applicability of the entire framework. For that, we are using the same traffic monitoring IoT application to generate the data for the application and publish the generated data to different IoT middleware platforms to compute performance metrics. Hence, the purpose of the section is three-fold. In the rest of this section, we will first discuss a case study based evaluation to validate the proposed IoTSySML model.To this end, we have modelled a traffic monitoring IoT application based on real-world scenarios. For the second part of the evaluation, we are running in-depth evaluations by designing experiments to validate the capability of the entire framework as well as the situation transition model, which is a core part of the framework. Finally, since our aim is to use the generated data to evaluate the performance of IoT middleware platforms, we will show that our framework can be used to compare different IoT middleware platforms based on defined metrics. We will discuss the performance metrics in the subsequent sections.

### 6.1. Case Study: Modelling IoT Traffic Monitoring Scenario

In this section, to exemplify the proposed IoTSySML model, we apply the modelling to a traffic monitoring IoT application to validate our modelling approach. We first present a brief overview of how a real-world IoT application scenario can be modelled in terms of context theory and situations, which is illustrated using a simplified block diagram in Figure 13.

The figure depicts the concepts followed by the example of the traffic monitoring IoT application scenario. We then discuss the modelling of our case study. Figure 13a illustrates the high-level concepts: sensors send the raw sensed to the context information layer. The contextual information combined with the sensory data enables reasoning about the situations. This is instantiated in Figure 13b where data from camera sensors combined with contextual information such as current location, and weather are used to define situations such as ‘low’, ‘moderate’, or ‘high’ traffic scenarios.

We now discuss the modelling of the case study. With reference to IoTSySML_X, we present a SySML diagram to represent our situation aware traffic monitoring IoT application (Figure 14). We are interested in identifying the situations an entity might be involved in. For instance, our entity here is *road_x*, a road segment, which can entail situations representing the traffic flow in road_x, which can be broken down to a situation of ‘low_traffic’, ‘moderate_traffic’ or ‘high_traffic’. Hence, we have a ‘involved in’ relationship between *road_x* and *low_traffic* situation. Situations are ‘associated With’ transitions, i.e., a situation of low_traffic might transition to moderate or high_traffic based on certain triggers. These have been captured by the *Transition* block in Figure 14. Situations can be associated with multiple transitions, shown by one-to-many multiplicity. The road_x entity has a *geoLocation* which represents context information. It also ‘has’ *CameraSensor* which ‘makes’ *Observations*. Figure 15 shows a JSON representation for the entity instance road_x. Camera sensors and Observations are associated with *Metadata* shown my one-to-many multiplicity in Figure 14. The observations have values ‘speed’, and ‘density’ represented by *ContextualParameters*, which are ‘associated with’ *Fuzzy Sets* for each observation. The fuzzy sets are in turn ‘described By’ FSI rules depicted by the *IF.AND.THEN* block. These rules are triggered to define the situations for our traffic scenario.

### 6.2. Experiment 1: Validating Situation Transition Model

The purpose of this section is to validate the situation transitions, which is a core component of SA-IoTDG. This experiment was conducted to evaluate the ability of the framework to support scenario based data generation. We evaluate this using our IoT traffic monitoring scenario, where we define three situations ($S_{1}$,$S_{2}$, and $S_{3}$) each allocated a probability for moving between states. For this experiment, we have assumed that the transition sequence is initiated with equally likely probabilities. Figure 16 illustrates the states involved in the transition model and the conditions which would trigger a situation to transition between the states. *‘Initial state’* is the state from which we begin our transition sequence. *‘Target State’* refers to the final state a situation transitions to, and *‘Transition state’* is any intermediate state, a situation reaches before moving to the target state. If we exemplify the transition model with our traffic monitoring scenario, we have three pre-defined situations: ‘low_traffic’, ‘moderate_traffic’, and ‘high_traffic’. We start the transition with ‘low_traffic’ and restrict the direct transition of ‘low_traffic’ to ‘high_traffic’ situation. To transition to ‘high_traffic’ situation, situations must first transition to ‘moderate_traffic’. Hence, our Initial State is ‘low_traffic’, the Target State would be ‘high_traffic’ and the Transition State would be ‘moderate_traffic’.

Situation transitions occur based on ‘time’ or based on allocated ‘Transition Probabilities (tPr)’ as illustrated in Figure 16. For a time based transition, users can specify the time during which a situation would occur, after which the next situation occurs as defined in the pre-defined sequence. For situation transitions based on transition probability, we initialize our Markov chain, define our start or initial state and allocate the probabilities for probability based transitions. We initialize our Markov model using an initial state probability vector given by:



$$P_{initial}=\left(p_{1},p_{2},p_{3}\right)$$



The initialization can be conducted in multiple ways: We can either assume a fixed starting point, or we can allocate probabilities for starting the Markov process. For this experiment, we assume equally likely probabilities for starting the sequence of transitions. However, this is configurable by users. Hence, they can change this depending on their application requirements. The situation transition module has been implemented in Python using the SciPy [71] library. We explain the situation transition model with the example of our traffic monitoring scenario. Since, we have three situations, ‘low_traffic’, ‘moderate_traffic’, and ‘high_traffic’, we define our initial state probability vector as:



$$p_{initial}=\left(0.33,0.33,0.33\right)$$



Next we also define our transition probability matrix as below:$$\begin{array}{c} \mathrm{tPr}=\left[\begin{array}{ccc} 0.5 & 0.5 & 0\\ 0.3 & 0.3 & 0.3\\ 0 & 0.5 & 0.5 \end{array}\right] \end{array}$$

Based on the above transition probability matrix tPr, and Figure 17, we can say that the situation ‘low_traffic’ can transition to ‘low_traffic’ and ‘moderate_traffic’ but not to ‘high_traffic’ since the transition probability from ‘low_traffic’ to ‘high_traffic’ is zero. Similarly, ‘high_traffic’ can move to ‘moderate_traffic’ and ‘high_traffic’ but not directly to ‘low_traffic’. In addition, ‘moderate_traffic’ can transition to all the situations: ‘low_traffic’, ‘moderate_traffic’ or ‘high_traffic’ as illustrated by the transition probability matrix in Figure 17. Figure 18 shows our Markov model based on our transition probability matrix.

#### Experimental Results for Situation Transitions

We have simulated the situations of a traffic monitoring scenario. To validate our situation transition model, we have implemented a random walk based on this model in Python. We initiated our transition sequence with a situation of ‘low_traffic’, and the generated random walk is shown below: 



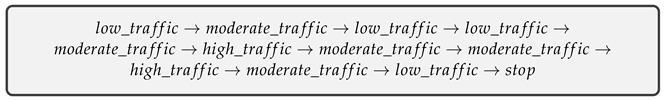



We evaluate our model by simulating a Markov sequence and visualize the result. As discussed earlier, to make the transition of situations realistic, we restricted our model to directly transition from situation ‘low_traffic’ to a situation of ‘high_traffic’ and vice-versa. However, ‘moderate_traffic’ can transition to both ‘low_traffic’ and ‘high_traffic’. This is validated in Figure 19, where the graph illustrates that there is no direct transition from situation ‘low_traffic’ to ‘high_traffic’. The sequence starts with situation ‘low_traffic’, then gradually transitions to ‘moderate_traffic’ followed by ‘high_traffic’. Hence, the model enables accurate transition from one situation to another.

### 6.3. Experiment 2: Evaluating the Capability of the Entire Framework, SA-IoTDG

In this section, we perform a set of experiments and analysis to validate the feasibility and applicability of the proposed framework. We evaluate SA-IoTDG based on the following aspects:mimicking the properties of IoT time series data i.e., we aimed to investigate if SA-IoTDG is capable of producing data that have similar properties of real-world data (Experiment 2.1 and 2.2)capturing different user requirements and generating data dynamically (Experiment 2.3)

We performed three sets experiments to achieve the above. We will first discuss the set-up used and then discuss the designed experiments.

#### 6.3.1. Experimental Setup and Metrics

To evaluate the proposed framework, the following set-up was used.

**Open Source Tools:** For this experiment, we used the IoT Data Simulator, implemented in Java, which allows both simulation and data replay. It also enables adding multiple devices and sending the device data to different target systems. We extended the tool to simulate the situations of an IoT application.

**SA-IoTDG:** The proposed framework for generating IoT data in line with defined situations is implemented using ReactJS, Node and Python. We have leveraged the Python FAST API framework to implement the situation description system using FSI. The situation transition module is conceptually modeled using Markov chains, and the simulations have been implemented using python and React JS. The framework is locally run using Visual Studio Code. It can be deployed using Docker as well.

**IoT managed cloud platforms:** The experiments were conducted on two different IoT middleware platforms, FIWARE [72] and AWS [73] to test their performance with different configurations.

We ran our experiments on a system running on MAC M1 chip, 16 GB RAM, with 1 TB SSD. During all experiments conducted, IoT data simulator were deployed using Docker. Similarly, FIWARE was also deployed as a docker container on the same system. For our experiments, we have developed a React application for simulating IoT application situations and generating relevant IoT data.

We will now discuss the metrics used to evaluate our implementations. To explore the similarities in the real and generated data by the framework, statistical parameters and autocorrelation function metrics were used.

To compute the performance of IoT middleware platforms, two main metrics that were considered are: (i). data ingestion rate and (ii). query response time. We explain these metrics below:*Data Ingestion Delay:* The difference in the time when the data is being generated and the time at which it is being inserted into the platform.*Query Response Time:* By query response time, we mean the time taken to execute a query after it is triggered on detection of an event.

The next two experiments have been designed to validate the applicability of SA-IoTDG based on the two factors discussed in the beginning of this subsection.

#### 6.3.2. Experiment 2.1: Investigating Data Distributions

To validate the data generation capability of our framework in keeping the significant features of real data, we identified the distribution of real and generated data. To achieve this, we compared the distribution of our generated traffic data with a real traffic dataset (Minnesota Department of Transportation). We found the best fitted distribution using the SciPy python library [71]. It is observed that both datasets follow a Burr distribution as the best fit based on the sumsquare_error parameter. The results are illustrated in Figure 20 which indicates that the distribution of the generated data is consistent with real-data. Hence, SA-IoTDG generates data following the same trend as real-data.

#### 6.3.3. Experiment 2.2: Exploring Similarity in Data Patterns

We examine the similarity between the two datasets using statistical methods. We compute statistical properties of the generated traffic data and compare with the real traffic dataset (Minnesota Department of Transportation) to evaluate if they follow similar trends and patterns. We generated high frequency time-series data with a sampling frequency of 1/5 s. We expect to observe multiple patterns in the data. Hence, we decompose our data to examine for trend and patterns. Figure 21 illustrates the components of the data. The top panels in Figure 21a,b shows the original data labelled as ‘data’. The remaining panels show the existing patterns in the data. From the figure, we can infer that both datasets exhibit a similar seasonal and trend component.

Since the generated data exhibits a trend component, we can say that the data are non-stationary. We confirm the non stationarity of the generated traffic data using autocorrelation function (ACF). ACF gives the correlation between the present values of a time-series data and the past values. The gradual decrease of the ACF in Figure 22a illustrates that the generated data are non-stationary. In order to fit the generated data in any model for further analysis, we need stationary data, so we transform the data to stationary by differencing them. Figure 22b shows the differenced data.

Additionally, we compute a statistical summary presented in Table 5. It can be observed that both datasets have very similar statistical properties, which can be used to infer that generated data follows properties similar to that of a real dataset.

#### 6.3.4. Experiment 2.3: Data Generation with Varying User Configuration

These experiments were designed to validated that SA-IoTDG is capable of capturing user specific requirements and generate data based on the specified configuration. We instantiate this using two examples. (1). Generating data by specifying a threshold time, and (2). specifying the data generation frequency in seconds. Figure 23a illustrates a configuration specified for 8 min (480 s) of data to be generated for each situation. We have modelled a traffic monitoring application as discussed, entailing situations of low_traffic, moderate_traffic, and high_traffic. Figure 23b illustrates data being generated for 8 min for each traffic situation.

Figure 24 illustrates a snapshot of data being generated by SA-IoTDG at a user configured frequency of 5 s. We also simulated data, mimicking a traffic monitoring application scenario. The data was aggregated at hour level. Figure 25 shows a sample of simulated data with the peaks’ representation data for a situation of high_traffic.

### 6.4. Experiment 3: Using SA-IoTDG to Conduct Performance Evaluation of IoT Middleware Platforms

This experiment was conducted to validate the frameworks’ capability to evaluate the performance of different IoT middleware platforms. To this end, we used the same traffic monitoring IoT application to generate the data for the application and published the generated data to two IoT middleware platforms AWS and FIWARE. The performance was analyzed based on metrics data ingestion delay and query response time as discussed in Section 6.3.1. Data ingestion delay was calculated by publishing the generated data to the two middleware platforms with a data ingestion rate (rate at which sensor data being inserted to a system per second) of 500 data points per second. Figure 26a compares the data ingestion delay of AWS and FIWARE. The result shows that the data ingestion delay is slightly more in FIWARE than that of AWS. Other metrics such as the query response time can also be computed by querying the ingested data on both platforms. Figure 26b illustrates how query response time varies with different data ingestion rates and varying complexity of queries. The experiment was conducted with data ingestion rates of 200, 500 and 1000 data per second, while varying the query complexity at the same time. The query complexity was varied by changing the number of joins and filter conditions. Since, the focus of the paper is not on evaluating the performance of the IoT middleware platforms, we did not provide a detailed technical evaluation. The aim is rather to show that the proposed framework can be used to conduct such performance evaluations.

## 7. Conclusions

In this paper, we proposed, implemented and evaluated SA-IoTDG, a novel framework to capture and simulate situation based IoT different application requirements and generate data. The proposed framework can: (1) define rules that can simulate real-world situations and stipulate the transitions of the defined situations, (2) generate IoT data based on the user specified configurations, and (3) use the framework to conduct performance evaluations of IoT middleware platforms. In existing works, we can currently generate data for single situations. The advantage of our proposed method is that, with our proposal, we can model multiple situations, specify the transition sequence and generate data. To the best of our knowledge, this is the first work to consider the transition of situations for generating IoT data. We illustrated that the proposed SA-IoTDG is capable of working with different configurations, and generates data with features similar to that of real data. Hence, SA-IoTDG is configurable and dynamic and is a step towards accelerating the development of a benchmark solution for IoT middleware platforms.

In this paper, our main focus was on generating IoT data relevant to situation specific IoT applications for evaluating IoT middleware platforms. Despite the contribution, there are some open issues that require further investigation. One of the interesting aspects we are planning to work on is:

*Data Extrapolation:* By data extrapolation, we mean generating data based on a sample data. In this paper, we have conducted experiments, where data was generated based on pre-defined rules by users. However, to provide more flexibility to users, and generate more realistic data, the framework should be able to generate data based on a sample data. To address this requirement, we plan to investigate and develop a mechanism to provide a dataset as seed and generate similar data for relevant IoT applications. Additionally, considering this work as a starting point, as future work, we plan to conduct further evaluations considering additional metrics to evaluate the performance of IoT middlewares.

## Figures and Tables

**Figure 1 sensors-23-00007-f001:**
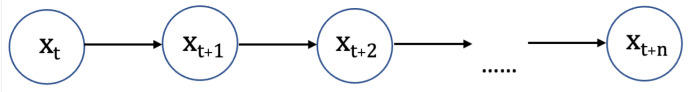
A Generic Markov Chain.

**Figure 2 sensors-23-00007-f002:**
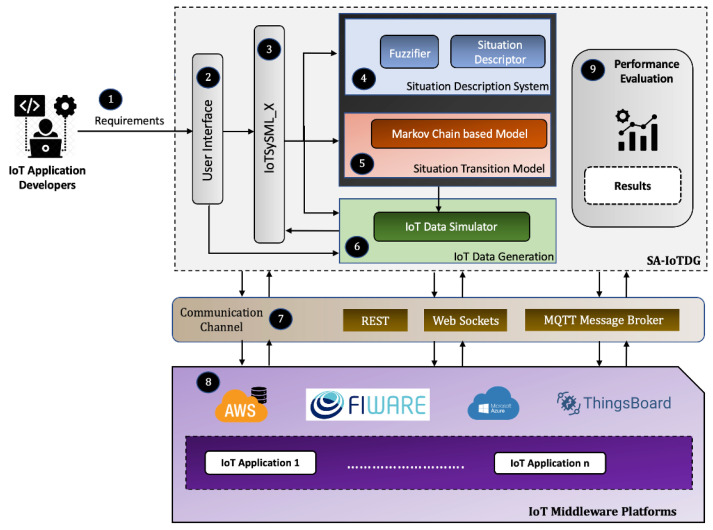
High-level System Overview.

**Figure 3 sensors-23-00007-f003:**
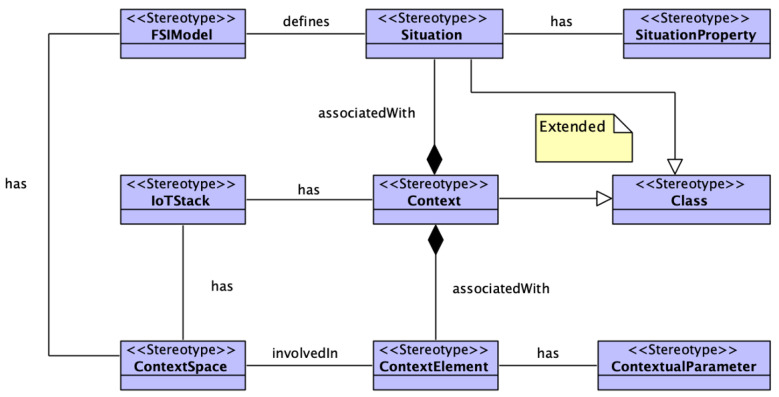
Extended class stereotypes.

**Figure 4 sensors-23-00007-f004:**
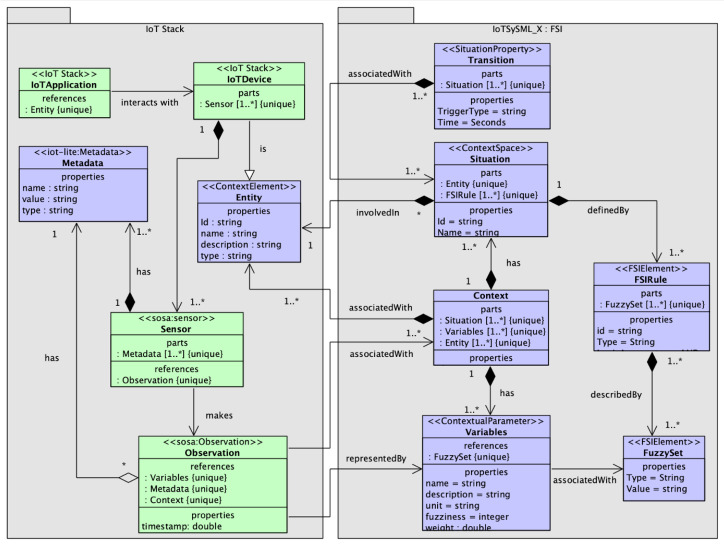
UML class diagram of the domain concepts.

**Figure 5 sensors-23-00007-f005:**
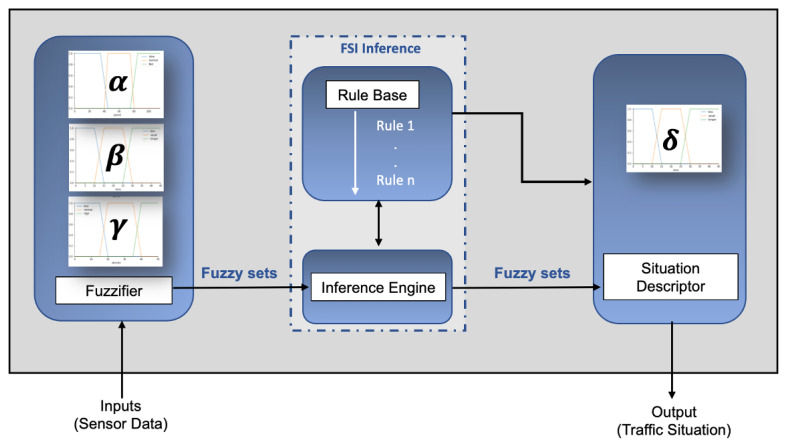
FSI based Situation Description System.

**Figure 6 sensors-23-00007-f006:**
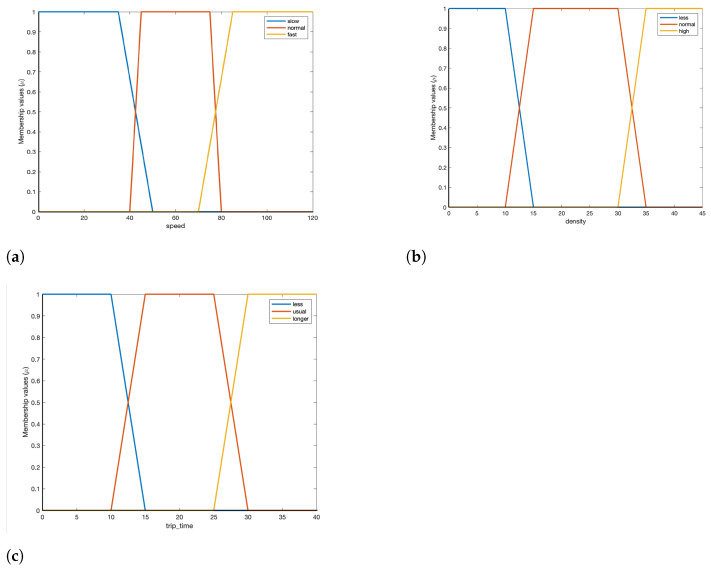
Fuzzy subsets for the input variables of a traffic monitoring application. (**a**) α. (**b**) β. (**c**) γ.

**Figure 7 sensors-23-00007-f007:**
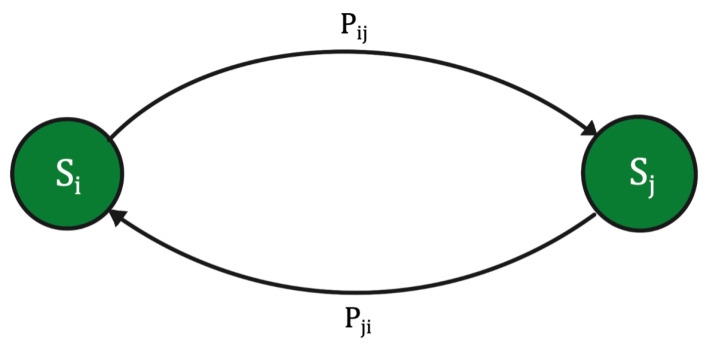
Two State Markov Chain.

**Figure 8 sensors-23-00007-f008:**
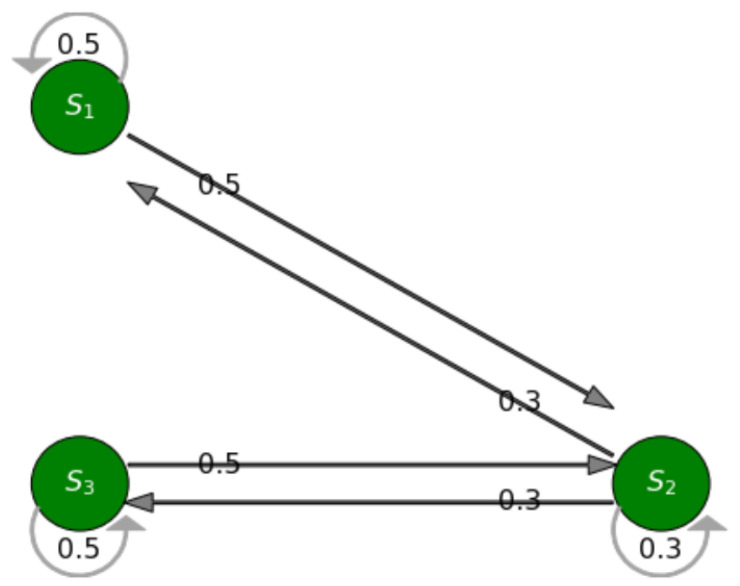
Three State Markov Chain.

**Figure 9 sensors-23-00007-f009:**
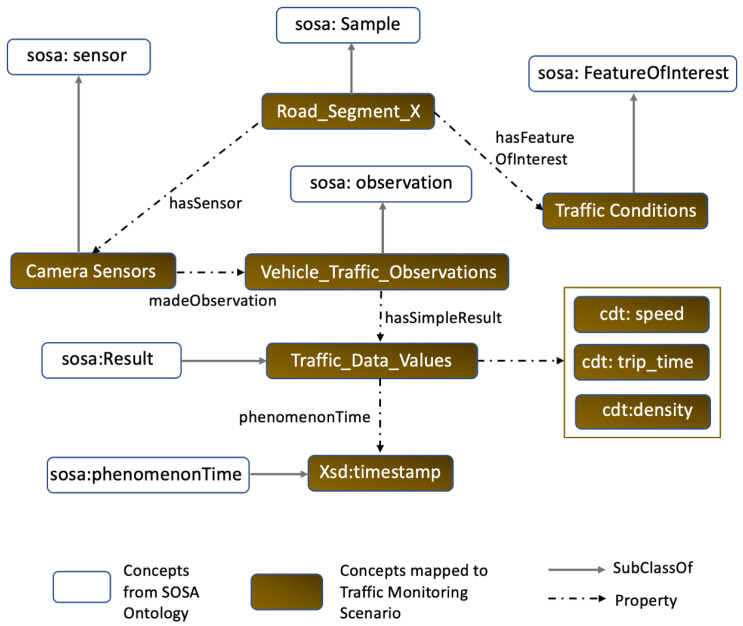
Illustration of the traffic sensor data.

**Figure 10 sensors-23-00007-f010:**
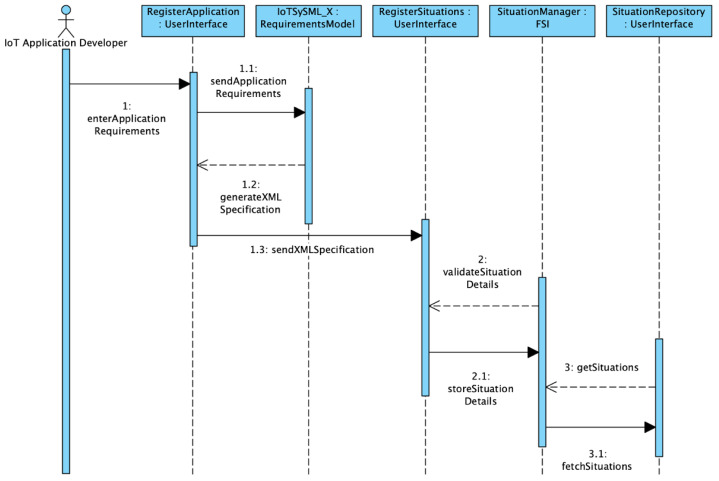
Registering situations.

**Figure 11 sensors-23-00007-f011:**
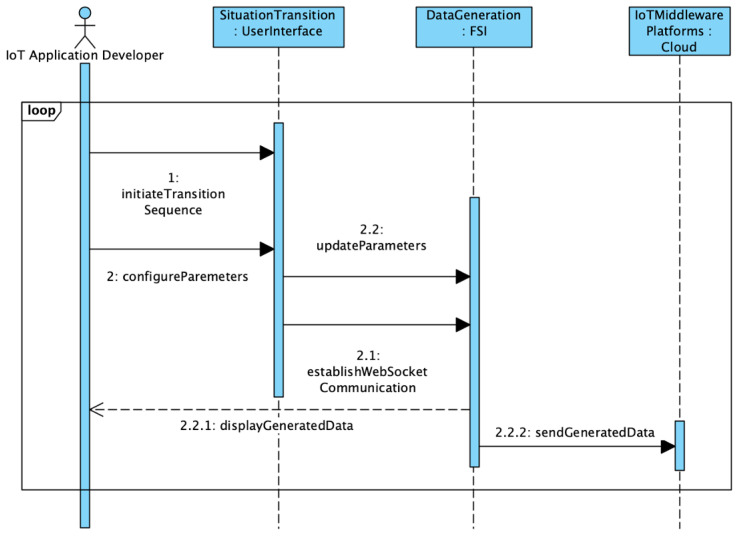
Running situation-based IoT data generation.

**Figure 12 sensors-23-00007-f012:**
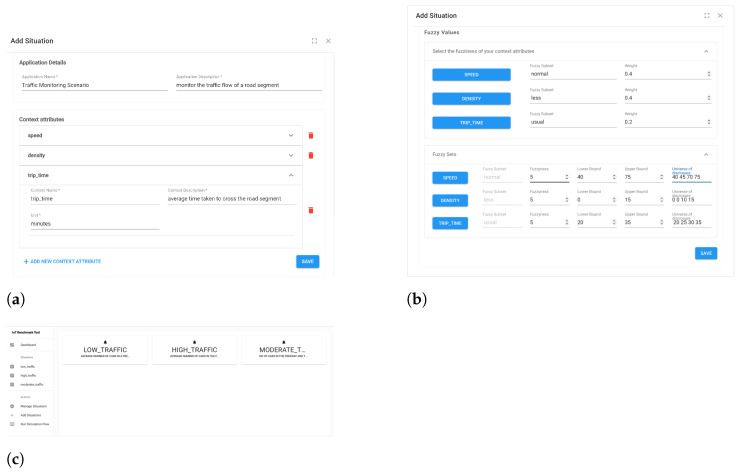
Snippets of the user interface to register situations and perform simulations: (**a**) register application details, (**b**) contextual information for registering situations, (**c**) dashboard view of registered situations.

**Figure 13 sensors-23-00007-f013:**
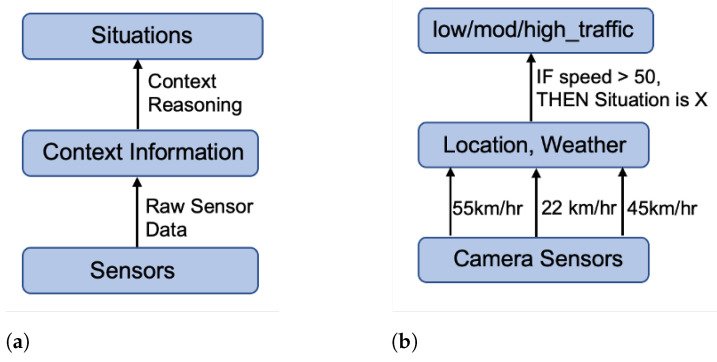
High-level overview of a traffic monitoring scenario: (**a**) concept, (**b**) example.

**Figure 14 sensors-23-00007-f014:**
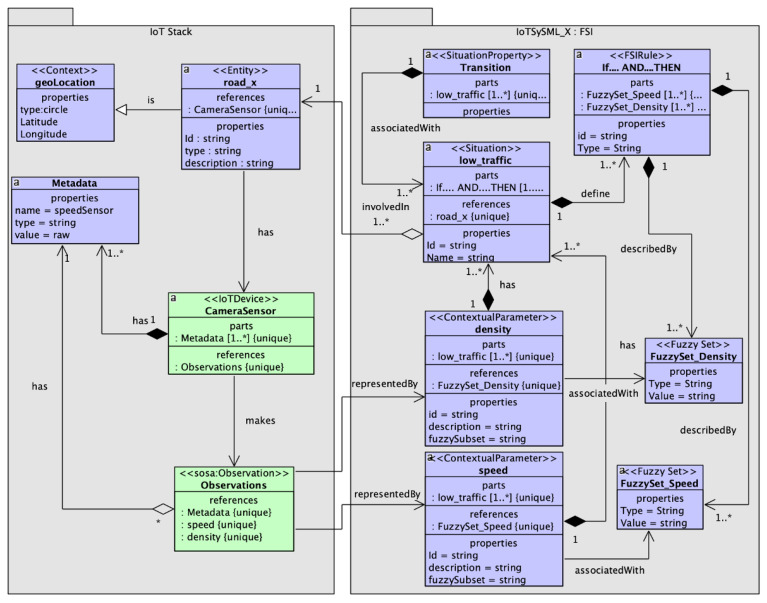
SySML diagram representing a Traffic Monitoring Scenario.

**Figure 15 sensors-23-00007-f015:**
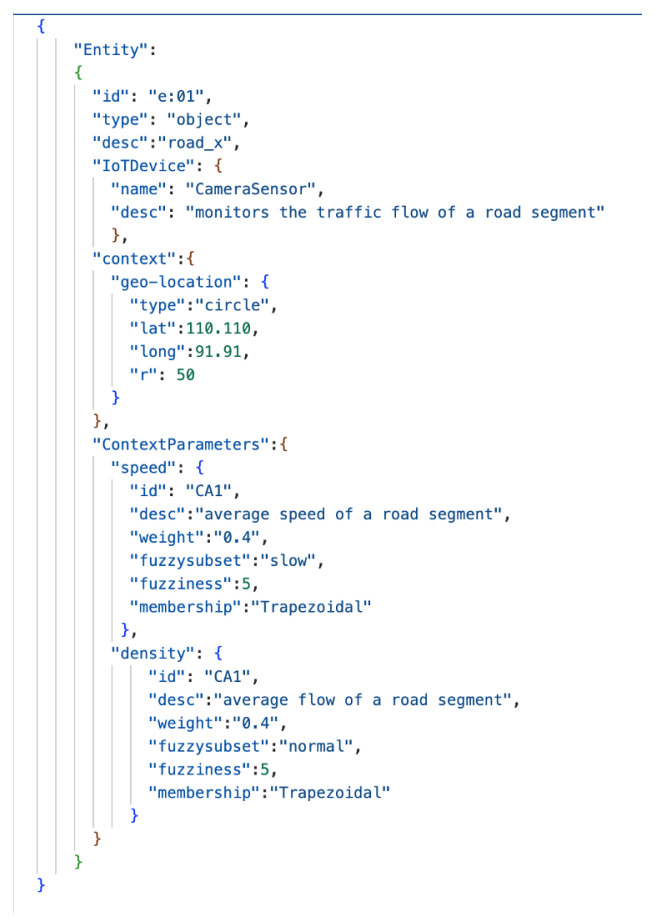
Sample JSON representation for an entity.

**Figure 16 sensors-23-00007-f016:**
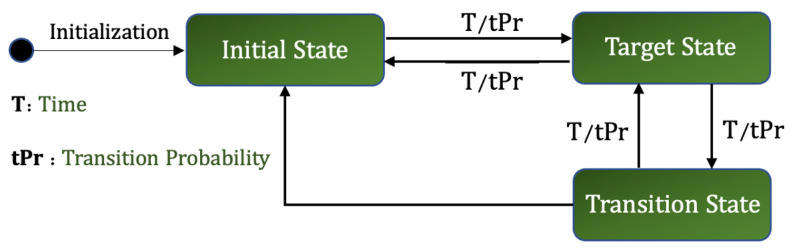
States and Conditions for Situation Transitions.

**Figure 17 sensors-23-00007-f017:**
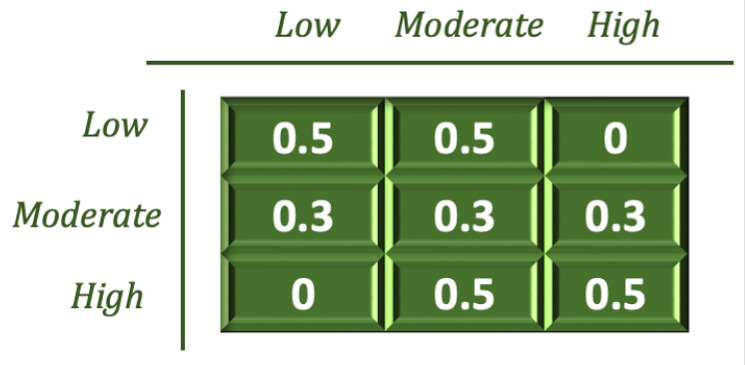
Transition Probability Model of Traffic Monitoring Scenario.

**Figure 18 sensors-23-00007-f018:**
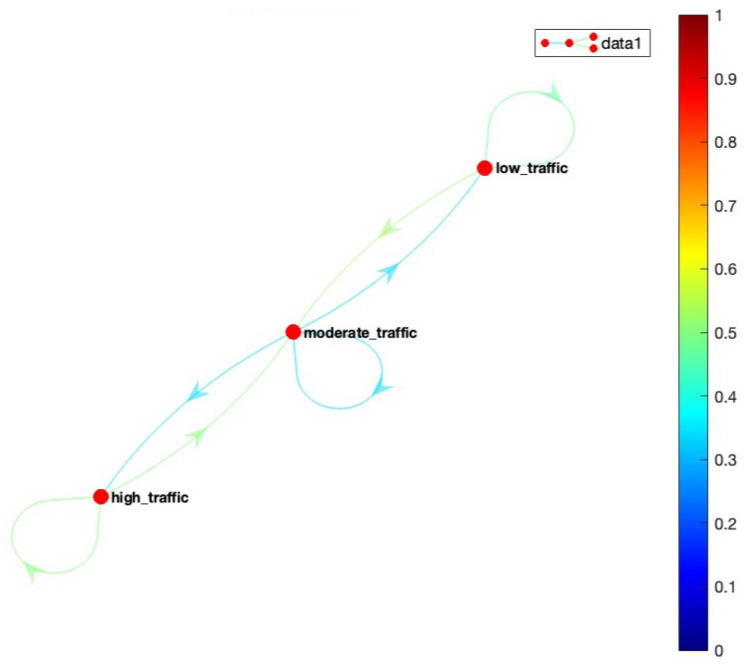
Markov Chain for traffic situations.

**Figure 19 sensors-23-00007-f019:**
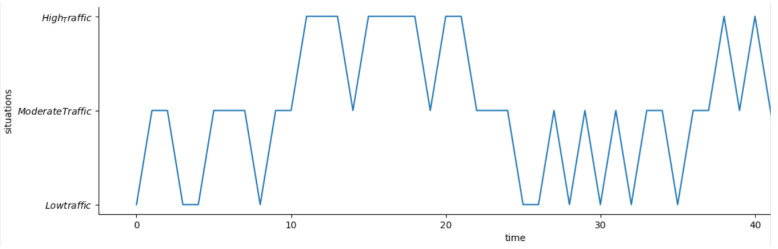
Markov Simulation for Situations in a Traffic Monitoring Scenario.

**Figure 20 sensors-23-00007-f020:**
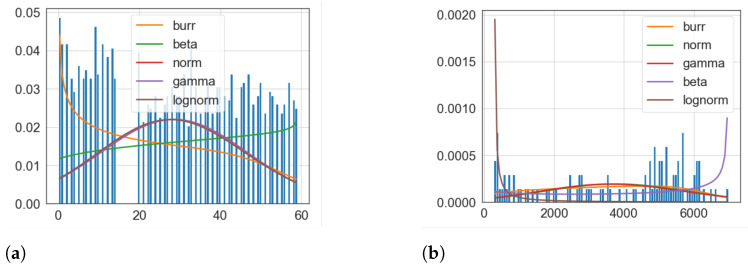
Investigating the best fit distribution. (**a**) Distribution of generated data. (**b**) Distribution of real data.

**Figure 21 sensors-23-00007-f021:**
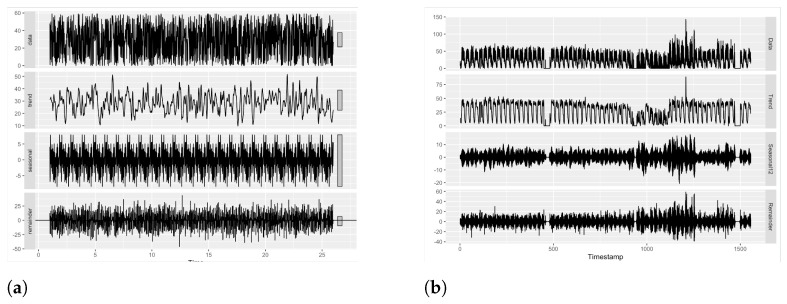
Comparing trends and patterns in generated and real data. (**a**) Components of generated data. (**b**) Components of real data.

**Figure 22 sensors-23-00007-f022:**
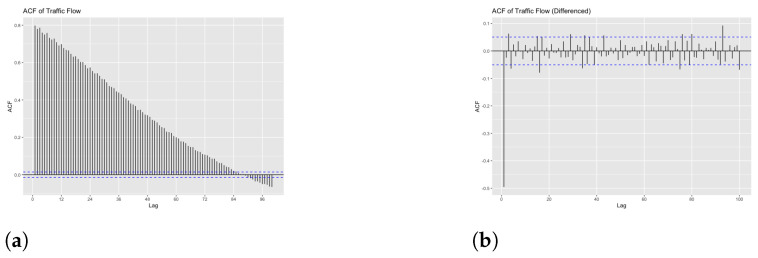
Autocorrelation functions of generated data. (**a**) ACF of generated traffic Data. (**b**) ACF of differenced data.

**Figure 23 sensors-23-00007-f023:**
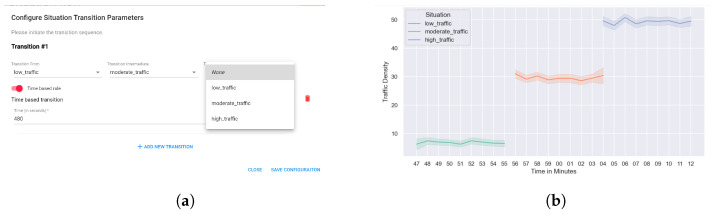
(**a**) Specifyingtime configuration. (**b**) Generated data based on configured time.

**Figure 24 sensors-23-00007-f024:**
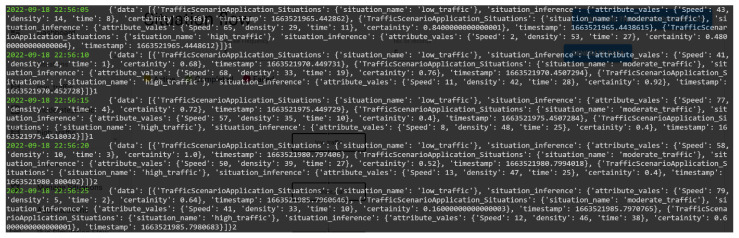
Data generated every 5 s.

**Figure 25 sensors-23-00007-f025:**
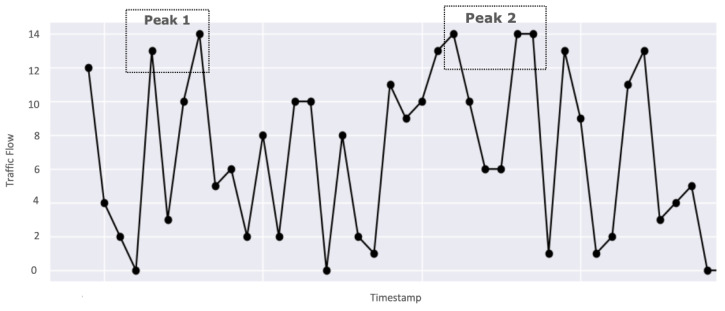
Visualizing generated data of a traffic monitoring scenario.

**Figure 26 sensors-23-00007-f026:**
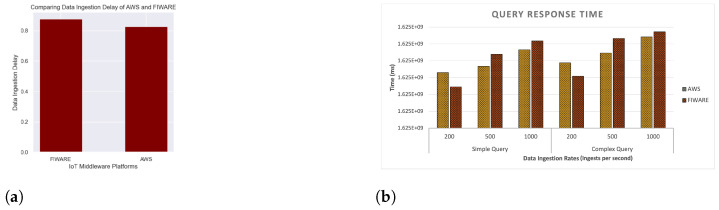
(**a**) Data ingestion delay. (**b**) Query response time of AWS and FIWARE [15].

**Table 1 sensors-23-00007-t001:** Summary of existing IoT benchmarks.

Paper	Benchmarking Targets	Evaluation Metrics	IoT Application Requirements	IoT Data Generation
[27]	IoT device		✗	✗
[35]	IoT Gateways	IoTps ${}^{1}$	✗	smart meter data
[36]	IoT Middleware Platform	IoTps ${}^{1}$	✗	smart meter data
[8]	IoT middleware platform	latency, throughput, CPU, memory utilization	✗	✗
[21]	IoT middleware platform	IPS ${}^{2}$, resource utilization	✗	✗
[23]	IoT middleware platform		✗	✗
[22]	IoT middleware platform	Contextual Queries	✗	✗
Our work	IoT middleware platform	IPS, DID ${}^{3}$, Query Response Time	✓	✓

^1^ Performance Metrics, ^2^ Ingests Per Second, ^3^ Data Ingestion Delay.

**Table 2 sensors-23-00007-t002:** Description of Relationships.

Relationship	Description	Base Class	Target Class
Composition	Objects that are associated with each other can not remain in the scope of a system without each other. The target class cannot exist without the base class i.e..if target class is deleted, base class gets deleted	Metadata	Sensor
		Sensor	IoT Device
		Entity	Situation
		Situation	Context
		Situation	Transition
		Variables	Context
		FuzzySet	FSIRule
Aggregation	Objects that are associated with each other can remain in the scope of a system without each other	Metadata	Sensor
		Metadata	Observation
Generalization	Represents an “is-a” relationship	Entity	IoTDevice
Association	Semantic relationship between classes representing a logical connection between them	IoTApplication	IoTDevice
		Observation	Variables
		Variables	FuzzySet

**Table 3 sensors-23-00007-t003:** FSI rule base for a traffic monitoring scenario.

	Input		Output
IF	AND	AND	THEN
A	B	C	X
speed is fast	density is less	trip_time is less	Low Traffic
speed is normal	density is less	trip_time is less	Low Traffic
speed is normal	density is normal	trip_time is usual	Moderate Traffic
speed is slow	density is high	trip_time is longer	High Traffic

**Table 4 sensors-23-00007-t004:** Example of calculating the confidence value for the occurrence of a situation ’Low Traffic’.

Input Variables	Values	Conditions	Weights	$\mu_{x}\left(i\right)$
speed	45	speed is normal	0.4	0.9
density	15	density is less	0.4	0.3
time	10	trip_time is less	0.2	0.5

**Table 5 sensors-23-00007-t005:** Comparing statistical summary of datasets.

Input Variables	Generated Data	Real Data
Mean	0.00	0.00
SD	18.34	17.34
1st Quartile	10.00	11.00
Median	29	30
3rd Quartile	40	44

## Data Availability

Not applicable.

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
