# Peer review of "Situation-Aware IoT Data Generation towards Performance Evaluation of IoT Middleware Platforms"

_sensors, 2022, doi:10.3390/s23010007_

Round 1
Reviewer 1 Report
A novel situation aware IoT data generation framework has been proposed in this paper. The paper is well written, and it contains sufficient explanation and extensive experiments were conducted to justify the performance of the research. I recommend accepting this paper in its current form.
Author Response
We thank reviewer 1 for this positive feedback and recommending our manuscript be accepted for publication.
Reviewer 2 Report
It’s also important if the author can summarize the advantages, and disadvantages of each related approach. Summarize the related works in a suitable table. Moreover, the research contribution in this paper must be made clear.
How to determine the fuzzy rules? The construction of fuzzy system is not explained well. How to prove/justify if the rule is valid or not? Is there any subjective judgement involved?
Please clarify whether the assumptions of the model are reasonable in reality.
The evaluation of the proposed model is too simple and is not comprehensive. More discussion, metrics, and comparison are necessary.
Author Response
Please kindly see the attachement.

Reviewer 3 Report
This is an interesting manuscript with application. All concepts are correct mathematically. I have some comments to improve the quality of the paper:
(1) Abstract should be rewritten seriously. It is better to mention the resulting advantages of the presented method in the abstract.
(2) Please provide a clear description about how the proposed method is better than previously developed methods.
(3) Summaries the advantages and limitations of the proposed method in practical applications.
(4)The conclusions should be extended with more future work.
Round 2
Reviewer 2 Report
The authors have made good revisions according to the comments of the anonymous reviewers.